# Body Composition Benefits Diminish One Year After a Resistance Training Regimen in Breast Cancer Patients, Although Improvements in Strength, Balance, and Mobility Persist

**DOI:** 10.3390/jfmk10020165

**Published:** 2025-05-09

**Authors:** Colin E. Champ, Jared Rosenberg, Chris Peluso, Christie Hilton, Rhyeli Krause, Alexander K. Diaz, David J. Carpenter

**Affiliations:** 1Allegheny Health Network Cancer Institute, Exercise Oncology & Resiliency Center, Pittsburgh, PA 15202, USA; christopher.peluso@ahn.org (C.P.); rhyeli.krause@ahn.org (R.K.); 2Department of Radiation Oncology, Allegheny Health Network, Pittsburgh, PA 15202, USA; 3Department of Exercise Science, SUNY Cortland, Cortland, NY 13045, USA; jared.rosenberg@cortland.edu; 4Department of Medical Oncology, Allegheny Health Network, Pittsburgh, PA 15202, USA; christie.hilton@ahn.org; 5Department of Radiation Oncology, Murray-Calloway County Hospital, Murray, KY 42071, USA; alexanderkdiaz@gmail.com; 6Department of Radiation Oncology, Wellstar Paulding Medical Center, Hiram, GA 30141, USA; davidcarpenter522@gmail.com

**Keywords:** breast cancer, resistance training, exercise science, hypertrophy, body composition

## Abstract

Objectives: Resistance training can improve body composition and physical function during and after breast cancer treatment and improve quality of life. It is unclear whether these changes persist once a person is no longer actively enrolled in a structured exercise regimen. Thus, we analyzed participants from the EXERT-BC protocol, assessing an intense exercise regimen in women with breast cancer at one year. Methods: All the participants were asked to undergo reassessment at one year. Current exercise habits, injuries, changes in medical history, body composition, handgrip strength, functional mobility and balance, and patient-reported quality of life were assessed. Pairwise comparison was performed via the paired t test. Results: Out of 40 initial participants, 33 returned for reevaluation, with 6 lost to follow-up and 1 with unrelated hospitalization. The median age was 57.8 years, and stage at diagnosis was 1. Weekly exercise was reported by 16 participants (48.5%), with 14 of the 16 following structured resistance training. Between completion of the EXERT-BC and one year follow-up, five women (15.2%) experienced musculoskeletal injuries, which inhibited their ability to exercise. Three women (9%), who were no longer exercising experienced orthopedic injuries requiring medical intervention. The significant reduction in percent body fat, total body fat, excess fat, and increases in muscle mass, resting metabolic rate, and whole-body phase angle dissipated at 1 year. Activity levels and quality of life were no longer significantly improved. However, strength, mobility, and balance remained significantly improved versus pre-exercise measurements, whether a participant was still engaged in exercise or not. Conclusions: After a 3-month dose-escalated resistance training regimen, exercise compliance was poor at one year. The anthropomorphic benefits of the regimen regressed by one year; however, the improvements in strength, balance, and mobility persisted.

## 1. Introduction

Body composition generally includes the percentage of muscle mass, adipose tissue, and bony mass within the human body. Optimal body composition generally results in a higher percentage of muscle mass versus adipose tissue. This metric is important in the cancer setting, as poor body composition is associated with worse survival in women with non-metastatic breast cancer [1]. Additionally, strength, which correlates with body composition and can be measured via the hand grip, is associated with improved quality of life and reduced mortality in individuals with breast cancer [2,3]. Strength and balance are of further importance in breast cancer patients as they often have a higher risk of falls and fractures secondary to both age and cancer treatment, and the side effects of breast cancer treatment can promote muscle loss, frailty, and balance dysfunction due to neuropathy [4]. Additionally, many receive anti-estrogen therapies that can promote bone density loss, raising fracture risk [5]. Thus, methods to offset these potential side effects of treatment, while improving strength and body composition, are needed.

Increased frailty and decreased strength and muscle mass are common sequelae from cancer treatment, leading to potential morbidity and mortality during survivorship [6,7,8]. Additionally, muscle mass loss may promote future comorbidities, including hormonal dysregulation, metabolic dysfunction, increased adipose tissue, and risk of orthopedic issues [9,10]. These factors would suggest that adequate methods to preserve, or even improve, muscle mass, strength, and physical function during and after cancer treatment should be an important component of breast cancer treatment and survivorship [11]. Resistance training protocols aimed at promoting neurologic and muscular adaptation while stimulating hypertrophy and strength enhancement may improve a range of general health- and treatment-related sequalae after the treatment of breast cancer [12].

Resistance training includes the lifting of weights at increasing thresholds of weight and intensity to promote physiologic accommodation through alterations in tempo, frequency, time under tension, rest time, and the type of exercises. While the general introductory recommendations for cancer patients include two sessions per week of at least two sets of 8–15 repetitions of intensity at least 60% of one repetition max, the long-term recommendations are less clear [13]. Additionally, these recommendations have been questioned as more optimal approaches may be necessary and supported through data in noncancer settings [12]. A large number of studies have assessed resistance training in the breast cancer setting, but doses may be inadequate [14]. Additionally, while studies have revealed similar exercise adherence during and after cancer treatment, long-term adherence in the breast cancer setting is less clear, and some data would suggest adherence strongly correlates with baseline activity levels [15,16].

Exercise, and specifically resistance training utilizing an effective dose that surpasses a threshold to promote muscular and neurologic adaptation, can improve body composition, strength, physical function, and metabolism due to the cellular and physiologic changes [12]. As these factors correlate with both improved outcomes and quality of life during and after the treatment for breast cancer, efforts are underway to incorporate resistance training as part of the standard of care in the treatment of breast cancer.

Along these lines, our group has previously shown significant changes in body composition, including decreases in adipose tissue and increases in muscle mass, along with improvements in mobility, strength, and balance after a three-month dose-escalated exercise regimen consisting of intense and high-volume resistance training [10]. However, it is unclear whether these metrics persist over the long term, especially as many participants no longer train in a structured or supervised fashion upon completion of the three-month exercise program. Thus, we performed a reanalysis of all the participants one year after the regimen, analyzing strength, balance, quality of life, and anthropomorphic variables associated with exercise and breast cancer treatment outcome. The aim of this study was to reanalyze the participants from the EXERT-BC protocol, assessing an intense exercise regimen in women with breast cancer at one year after completion to assess body composition, strength and mobility metrics, and exercise adherence.

## 2. Materials and Methods

### 2.1. Participants

EXERT-BC was an institutional review board-approved (Allegheny Health Network Review Board) single-arm unblinded phase I trial of a single exercise cohort (protocol 2022-269-SG) registered at ClinicalTrials.gov (NCT05747209) and conducted in accordance with the TREND guidelines as previously described. No matching occurred during enrollment. A consent form was distributed to all the participants and signed. Women aged 20–89 years with a breast cancer or ductal carcinoma in situ (DCIS) diagnosis were eligible and were referred to the Exercise Oncology Center via their oncologist or surgeon. The participants were evaluated by a practitioner with dual medical doctor (MD) and Certified Strength and Conditioning Specialist (CSCS) certification and deemed eligible to participate in a group exercise session. Individuals were required to be able to get up and down from the ground and perform a body weight squat to 90°. Individuals on chemotherapy were not eligible, but radiation therapy and/or non-cytotoxic systemic therapy was allowed during the protocol. Individuals with severe arthritic, joint, musculoskeletal, or cardiovascular comorbidities interfering with the ability to safely resistance train were excluded from the protocol. Informed consent was obtained from all the subjects. In total, 40 women with a history of breast cancer not receiving cytotoxic chemotherapy were enrolled and completed the protocol between 30 December 2022 and 13 August 2024.

### 2.2. Training Program

The exercise regimen consisted of thrice-weekly intense resistance training for three months utilizing compounding movements and linear progression, along with an adequate volume, with the goal of increasing strength and mobility and promoting hypertrophy (ten sets of ten repetitions per movement). Individuals trained in groups, and each session took 45–60 min and was under the supervision of Certified Strength and Conditioning Specialists at the gym at our Exercise Oncology Center to enhance safety and compliance. Exercises generally progressed from compound movements, like split squat and hex bar deadlift, utilizing linear progression, to more isolated movements at hypertrophy repetition schemes (3–4 sets of 10 repetitions). The participants underwent a 2-week ramp-up period utilizing the same exercises, but the sets were gradually increased. An example of the first month of the exercise regimen can be seen in Table 1. Before the regimen and afterward, individuals underwent workups for body composition, muscle mass, fat mass, fat-free mass, and resting metabolic rate with bioimpedance analysis (InBody Co., Seoul, Republic of Korea) and ultrasound utilizing the Jackson and Pollock calculation (BodyMetrix, Brentwood, CA, USA). Balance and mobility were evaluated via a seven-movement functional movement screen (FMS) and Y-balance test, quality of life via EQ-5D-5L, activity levels via Godin Leisure-Time Exercise Questionnaires, and strength via hand grip strength. All the testing occurred at the adjacent Exercise Oncology Lab. The initial hypothesis and primary objective of the protocol was to assess whether this regimen could be adhered to and safely performed in women who underwent treatment for breast cancer, while potentially increasing strength, defined as load lifted; improving muscle mass; and increasing strength and mobility.

Significant improvements were measured at the initial planned assessment, with all the participants completing the three-month exercise regimen [10]. The participants then segued to exercise at home or at a local facility. The reassessment included current exercise adherence, body composition, quality of life, balance, and mobility. The primary endpoints included changes in body composition (i.e., percent body fat and muscle mass), phase angle via bioimpedance analysis, FMS and Y-balance test, resting metabolic rate (RMR), quality of life, post-intervention activity levels, and strength. Further details on these assessments have been previously described [10].

### 2.3. Statistical Analysis

Anthropometric, metabolic, fitness, and quality-of-life measurements were the secondary objectives and were analyzed as continuous variables. After assessing for normality defined as Shapiro–Wilk *p* > 0.05 confirmed via a visual inspection of histogram and dot plot distributions, changes from the baseline were assessed using the paired *t*-test and the Wilcoxon signed-rank test for parametric and non-parametric parameters, respectively. Missing data, including one set of hand grip strength and FMS in an injured participant, were excluded from corresponding pair-wise analyses. All the statistical analyses were performed using R version 4.1.2 (R Project for Statistical Computing, Vienna, Austria).

## 3. Results

In total, 33 participants returned at one year for reevaluation; the remaining 7 were alive, but, of them, 6 were lost to follow-up and 1 was unable to follow up due to unrelated hospitalization (Figure 1). The median patient age at time of reevaluation was 57.75 years, with 15 patients (45.5%) over 60 years (range, 28–76 years; Table 2). Concurrent oncologic therapies during the reassessment included anti-estrogen therapies (81.8%), and non-hormonal systemic therapies (15.2%; abemaciclib *n* = 5.). All the women underwent breast surgery, with 11 (33%) undergoing mastectomy with reconstruction; 4 (12.1%) underwent axillary lymph node dissection; and 18 underwent (54.5%) sentinel lymph node biopsy. Thirty individuals (90.1%) received radiation therapy (RT) during the exercise regimen. The median stage at diagnosis was stage 1.

Out of the 33 women evaluated at one year, 16 (48.5%) described continued weekly exercise. All of these individuals were injury free at one year, and 14 of the 16 remained on a structured resistance training program. Between the completion of EXERT-BC and the one-year follow-up, five women (15.2%) experienced exercise-related musculoskeletal injuries that inhibited their ability to exercise. Three women (9% total, or 17.5% of those no longer exercising) who were not exercising experienced orthopedic injuries unrelated to activity and requiring medical intervention, including a meniscus tear, a gastrocnemius sprain, and nonspecific knee pain. Four women (12.1%) stopped exercise due to surgeries related to their breast cancer. Of the six women on the initial EXERT-BC study with a history of exercise, all continued exercise at one year.

The pre- and post-intervention and one-year assessment values for body composition, strength and mobility, and quality-of-life parameters are provided in Table 3. Generally, the metabolic and anthropomorphic factors that were improved on initial evaluation were no longer significant at one year (Figure 2). No significant change was seen in weight at any time point. While significant reductions were seen in percent body fat at the completion of the exercise regimen, the total body fat, excess fat, and significant increases in muscle mass amount and percent, resting metabolic rate, and whole-body phase angle changes were no longer significant at 1 year. The Godin measurement of activity levels and the quality of life via EQ5D were no longer significantly changed and regressed to baseline levels.

Improvements in strength via handgrip, mobility via FMS, and balance via Y-balance test, however, remained significantly improved one year after the exercise intervention compared to before the regimen (Figure 1). Grip strength remained significant versus pre-evaluation and similar to that at post-exercise analysis (right overhead from 19 kg [IQR 18–24] to 24.5 [IQR 20–28], *p* < 0.001, to 26 [IQR 21–29], *p* < 0.001 versus initial; right at waist from 19 kg [IQR 18–24] to 24.5 [IQR 20–28], *p* = 0.02, to 26 [IQR 21–29], *p* < 0.001 versus initial; left overhead from 20 kg [IQR 18–27] to 26 kg [IQR 22–31], *p* < 0.001; and left at waist from 20 kg [IQR 18–27] to 26 [IQR 24–30], *p* < 0.001). The FMS increased from 9.0 [IQR 8.0–11.0] to 11.5 [IQR 10.0–13.3]. Balance testing on the right increased from 74.8 [IQR 67.0–79.6] to 84.1 [IQR 75.5–90.3], *p* = 0.003, and on the left, it increased from 72.7 [IQR 64.3–81.7] to 84.1 [IQR 75.5–90.3], *p* = 0.005.

The above variables were analyzed in the 17 individuals no longer exercising at one year versus the 16 individuals still exercising. Body composition, including fat mass, muscle mass, percent body fat, percent muscle mass, and fat-free mass changes regressed at one year in both subgroups and were no longer significant at one year. The phase angle and resting metabolic rate were no longer significantly changed, while balance, FMS, and hand grip strength remained significant in both subgroups regardless of present exercise.

Lastly, only 2 of the 17 individuals (12%) who were no longer exercising had a history of exercise prior to initiation of the protocol, while 6 of 16 individuals (38%) who were still exercising had been exercising prior to the initial exercise protocol.

## 4. Discussion

The primary purpose of this study was to assess body composition, strength metrics, and exercise adherence in individuals by reanalyzing the participants from the EXERT-BC protocol, thereby assessing the effects of an intense exercise regimen in women with breast cancer at one year after completion. A novel dose-escalated resistance training regimen employing high-intensity compound exercises in women treated for breast cancer revealed significant metabolic, anthropomorphic, and physical changes after 3 months. Most participants in this protocol did not continue an exercise regimen after completion, with just over 40% continuing a structured exercise program. A considerable portion of women experienced injuries after the protocol (25%), and nearly half of these were orthopedic injuries in individuals who were not physically active and no longer exercising. By one year, the initial positive anthropomorphic changes seen after the protocol were no longer present; however, the improvement in strength, mobility, and balance persisted at one year. Interestingly, the changes in strength, mobility, and balance were preserved regardless of whether an individual continued to exercise, likely due to neurologic adaptation.

Our findings demonstrate that, in a population of breast cancer patients, many of the functional benefits gained during a three-month dose-escalated resistance training regimen persisted at one year. Because improvements in muscle mass and body composition generally require significant effort and quantity of resistance training to reach a threshold and promote muscular adaptation, it is not necessarily surprising that the anthropomorphic improvements did not persist at a one year. It is surprising, however, that whether individuals were exercising or not, they did not maintain the improvements in body composition. The latter finding illustrates the importance of a supervised exercise program in maintaining muscle mass, likely due to the greater accountability in training volume, intensity, and progression that expert supervision often provides. Additionally, the group sessions with other participants of varying ages and skillsets likely helped motivate individuals to exercise intensely.

Intense resistance training utilizing heavy weights with loads lifted at 1–10 maximum repetition and at full range of motion at an adequate threshold can promote neurologic adaptation, which ultimately leads to improved strength, balance, and mobility [17,18,19]. Our data suggest that this neurologic adaptation persists for a prolonged period of time and beyond anthropomorphic changes. The persistence of the physical benefits of the regimen, including lower fall risk, improved balance, increased strength on hand grip, and improved functional mobility one year out, irrespective of subsequent exercise status, is highly encouraging and highlights that resistance training at an adequate threshold for 3 months can provide durable improvements in physical functioning. By extension, these enhanced physical attributes are likely to reduce morbidity and mortality associated with frailness, including falls and fractures. In other words, only three months of intense training might lead to at least a year’s worth of protection.

Resistance training, particularly utilizing compound movements and heavy weights to engage a substantial number of muscle fibers, can endorse adaptations in motor unit function from spinal cord output. This adaptation includes decreases in the motor unit recruitment threshold and increases in the motor unit discharge rate [20]. During our resistance training regimen, the participants were pushed to near volitional failure (i.e., 0–1 repetitions in reserve or a relative perceived exertion > 8), which may have resulted in increased muscle adaptation due to augmented rates of neurological firing [21], and an increase in type IIA muscle fibers, which can improve functional performance, particularly in an elderly population that is at risk of the loss of this muscle fiber [22].

Hand grip strength is associated with quality of life in breast cancer patients and cancer mortality [2,3]. Our data show that even three months of intense exercise can promote improvements in hand grip strength for up to one year after a workout regimen, potentially impacting both survival and quality of life. While longer follow-up and further studies are required, our data did show that the EQ5D scores regressed back to baseline at one year.

The initial compliance with the protocol was excellent, but exercise compliance was poor overall. While 98% of women continued an exercise regimen at the completion of the protocol [10], this number dropped to below 50% by one year. Additionally, most individuals exercising at one year had been exercising prior to the protocol. Thus, dropout rates were the highest for novices. These data are consistent with a large metanalysis analyzing 27 randomized controlled trials analyzing an exercise intervention in adults with cancer, which revealed that several factors are associated with exercise adherence, including contact with other participants, a supervised element, social support, and an exercise prescription [23]. Additionally, this may illustrate financial implications as the initial protocol was provided without a fee as part of the research study.

After three months on the protocol, many participants saw large changes in their overall health, functional performance, and quality of life. However, these improvements were insufficient to promote maintenance of the exercise regimen once these individuals completed the program and were no longer supervised by our team of certified strength and conditioning specialists (CSCSs), consisting of physicians and exercise physiologists. These results further reveal that, if medicine, as a field, desires patient adherence to any exercise program—an intervention that demonstrably improves overall health—then in-person regimens equipped with trained personnel are required.

Lastly, while concerns of injury rates have limited the intensity of exercise programs in the past [12,15], the high rate of orthopedic injuries in individuals not exercising was notable. With prior protocols showing lower injury rates during intense exercise protocols in these individuals, this may provide comfort that an intense structured resistance training program is not only safe but may lead to fewer orthopedic injuries than the avoidance of exercise. Along these lines, it should be noted that, while this regimen was observed by expert personnel, the equipment used was mostly inexpensive free-weights, allowing this regimen to be utilized in a range of settings.

### Limitations and Practical Applications

Several limitations should be considered. The above protocol was not a randomized trial and selection bias was likely; thus, these limitations must be considered. Overall, the numbers were small, inhibiting further analyses of competing variables and factors that may have impacted our results. However, this may provide insight for further areas of research, as exercise dropout rates were high in these potentially motivated individuals. A large, randomized trial with a long follow-up could help confirm some of the above findings. The results further stress that resistance training with mostly inexpensive and widely available weights can provide lasting improvements for breast cancer patients during survivorship. Additionally, many participants did not have a history of resistance training, yet they experienced significant improvements with a low rate of injury, further illustrating that this regimen can be utilized broadly in practice.

## 5. Conclusions

In summary, breast cancer patients on an intense resistance training protocol with good adherence experienced significant improvements in body composition and balance, mobility, and strength. However, by one year, exercise adherence was poor, and the benefits in body composition regressed. The physical changes, including strength, balance, and functional mobility, remained significant at one year. These findings are encouraging as the functional changes from an intense resistance training regimen may persist long term. For women undergoing treatment for breast cancer, intense exercise can be considered as an adjunct to treatment to offset the potential side effects of treatment and improve overall physical function. Additionally, efforts to provide these exercise interventions long term may help with compliance throughout survivorship.

## Figures and Tables

**Figure 1 jfmk-10-00165-f001:**
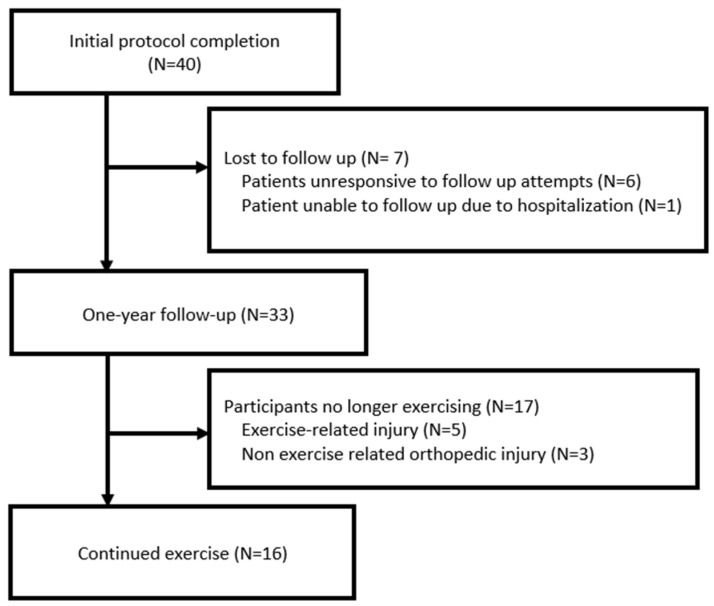
Consort diagram.

**Figure 2 jfmk-10-00165-f002:**
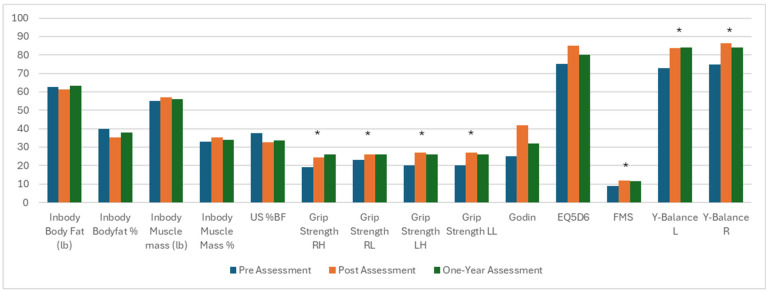
Metabolic and physical changes from prior to the exercise regimen to reassessment at 3 and 12 months. Significant changes between pre-exercise and one year assessment noted with an asterisk. US = ultrasound, RH = right overhead, RL = right at waist, LH = left overhead, LL = left at waist, FMS = functional movement screen, EQ5D = EuroQol 5 Dimension Questionnaire, Y-Balance L = balance left, Y-Balance R = balance right. * = statistically significant.

**Table 1 jfmk-10-00165-t001:** Month 1 of the exercise program.

Monday				Wednesday				Friday			
	EXERCISE	Sets	Reps		EXERCISE	Sets	Reps		EXERCISE	Sets	Reps
A1	SPLIT SQUAT	4	8	A1	GOBLET SQUAT	4	8	A1	HEX BAR DEADLIFT	4	8
A2	SIDE PLANKS	3	10	A2	BAND PULL APARTS	3	10	A2	TRX ROW	3	10
B1	BIRD DOG ROW	3	10	B1	HIP THRUST (12” plyo box)	3	10	B1	BOX STEP UP	3	10
B2	1-LEG GLUTE BRIDGE	3	10	B2	INCLINE DB PRESS	3	10	B2	PUSH UP	3	10
C1	1/2 KNEELING SHOULDER PRESS	3	10	C1	DB SKULL CRUSHERS	3	10	C1	SUITCASE CARRY	3	10
C2	BICEP DB CURLS	3	10	C2	DB EXTERNAL ROTATION ON KNEE	3	10	C2	2-LEG CALF RAISES	3	10

DB = dumbbell, Reps = repetitions.

**Table 2 jfmk-10-00165-t002:** Patient demographics and breast cancer characteristics.

	N	%
Median Age at Enrollment (Interquartile range)	57.8	(50–63)
Prior History of Exercise	9	27
Breast Cancer Stage		
Ductal carcinoma in situ	4	12
Early stage	23	70
Locally advanced	4	12
Locoregional recurrence	1	3
Distant metastases present	1	3
Lymphedema at Time of Enrollment	7	21
Concurrent Receipt of Anti-Estrogen Therapy	17	52
Concurrent Receipt of Non-Hormonal Systemic Therapy *	6	18
Concurrent Radiotherapy	19	58

* Concurrent therapies: abemaciclib, trastuzumab deruxtecan, trastuzumab emtansine, trastuzumab, pembrolizumab.

**Table 3 jfmk-10-00165-t003:** Body composition, physical function, and quality-of-life parameters before and after exercise intervention.

	Baseline (Median [IQR])	Post-Intervention (Median [IQR])	*p* Value Before to After	1 Year (Median [IQR])	*p* Value Before 1 Year	*p* Value After 1 Year
Height (inch)	65 (63–66)		N/A		N/A	N/A
Weight (initial)	167.8 (150.2–187.6)	160.3 (143.1–188.7)	0.88	163 (143–187.6)	0.76	0.96
InBody Body Fat (lb)	62.7 (47.6–80.6)	61.2 (43–78.7)	0.002	63.4 (45.5–73.5)	0.83	0.79
InBody Body Fat (%)	39.8 (33.7–42.3)	35.4 (30.5–42.8)	0.002	38 (31.2–42)	0.36	0.29
US Body Fat (%)	37.7 (33–39.2)	32.5 (29.1–37.5)	<0.001	33.7 (30.2–37.2)	0.36	0.29
US Essential Fat (lb)	54.8 (46.4–58.8)	55.0 (40.1–59.8)	0.88	55.4 (43.2–59.6)	0.77	0.75
US Excess Fat (%)	6.6 (0–16.4)	1.6 (0–10.6)	0.005	0 (0–6.63)	0.16	0.98
InBody Muscle Mass (lb)	55.1 (51.6–61.3)	57.1 (52.3–63.1)	<0.001	56 (52.7–60)	0.81	0.59
InBody Muscle Mass (%)	33.1 (31–35.8)	35.2 (31.2–38.0)	<0.001	34 (31.5–37.8)	0.52	0.57
InBody Fat-Free Mass (lb)	101.4 (94.4–111.6)	106.5 (95.9–113.3)	0.09	103.6 (96.3–109.1)	0.56	0.66
InBody RMR (kcal)	1367 (1298–1456)	1414 (1309–1480)	0.02	1385 (1314–1438)	0.95	0.67
US RMR (kcal)	1429 (1325–1493)	1432 (1353–1540)	<0.001	1442 (1349–1502)	0.38	0.70
Bone Mineral Content (kg)	6.22 (5.58–6.55)	6.3 (5.6–6.7)	0.15	6.3 (5.7–6.6)	0.88	0.94
Whole-Body Phase Angle (degrees)	4.9 (4.6–5.2)	5.0 (4.8–5.5)	<0.001	5.1 (4.7–5.5)	0.59	0.36
Right Grip Strength, Overhead (kg)	19 (18–24)	24.5 (20–28)	<0.001	26 (21–29)	<0.001	0.21
Right Grip Strength, at Waist (kg)	23 (19–28)	26 (22.6–29)	0.006	26 (24–28)	0.02	0.69
Left Grip Strength, Overhead (kg)	20 (16–23)	27 (22–28.3)	<0.001	26 (22–31)	<0.001	0.30
Left Grip Strength, at Waist (kg)	20 (18–27)	27 (22.8–29.3)	<0.001	26 (24–30)	<0.001	0.79
Functional Mobility Screen Score	9.0 (8–11)	12.0 (10–14)	<0.001	11.5 (10–13.3)	<0.001	0.84
Y-Balance Test Score, Left Side	72.7 (64.3–81.7)	83.7 (78–92)	<0.001	84.1 (75.5–90.3)	0.003	0.53
Y-Balance Test Score, Right Side	74.8 (67.0–79.6)	86.2 (76.9–94.2)	<0.001	83.9 (75–91.7)	0.005	0.43
EQ5D1 Score	5 (5–5)	5 (5–5)	N/A	5 (5–5)	N/A	N/A
EQ5D2 Score	5 (5–5)	5 (5–5)	N/A	5 (5–5)	N/A	N/A
EQ5D3 Score	5 (5–5)	5 (5–5)	N/A	5 (5–5)	N/A	N/A
EQ5D4 Score	4 (4–5)	4 (4–5)	N/A	4 (4–5)	N/A	N/A
EQ5D5 Score	5 (4–5)	5 (5–5)	N/A	5 (4–5)	N/A	N/A
EQ5D6 Score	75 (60–90)	85 (75–95)	0.02	80 (70–90)	0.31	0.15
Godin Questionnaire Score	25 (8–35)	42 (33–54)	<0.001	32 (19–47)	0.10	0.04

Kg = kilogram, US = ultrasound, kcal = kilocalories, RMR = resting metabolic rate, IQR = interquartile range, EQ5D = EuroQol 5 Dimension Questionnaire.

## Data Availability

The datasets generated and/or analyzed during the current study are not publicly available due to ongoing analysis and manuscript creation but are available from the corresponding author on reasonable request.

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
