# Peer review of "Body Composition Benefits Diminish One Year After a Resistance Training Regimen in Breast Cancer Patients, Although Improvements in Strength, Balance, and Mobility Persist"

_jfmk, 2025, doi:10.3390/jfmk10020165_

Round 1
Reviewer 1 Report
Comments and Suggestions for Authors
Dear authors,
your paper is valuable in its aims but some concerns have to be better addressed.
Methods have to be improved significantly.
In the methods section, a separate paragraph for statistical methods is required. You performed a pre/post study, but you should also specify which tests you used in case of parametric/non parametric variables.
Then, the design is not clear and it must be described both in the abstract and in the methods. Is it a retrospective study? Is it on a single cohort? Please specify and explain all these aspects.
As regards the results, many concerns arise. A regression is required to assess if some factors could affect the results post to 1 year. Overall, it is useful to investigate other factors which could have affected results: for example, did participants carry out sporting activities? Also this factors can effectively influence physical function, body and mineral composition and so on. This aspect should be described in the discussion, in order to make it more objective and to remember that sport can also modify a disease perception by the patients. To do that, I suggest the following reference which explain how sport affect the perception of another disease by the patients:
Notarnicola A, Farì G, Maccagnano G, Riondino A, Covelli I, Bianchi FP, Tafuri S, Piazzolla A, Moretti B. Teenagers’ perceptions of their scoliotic curves. an observational study of comparison between sports people and non- sports people. Muscles Ligaments Tendons J [Internet]. 2019;9(2):225-35.
I suggest to separate the limitations of this study, specifically describing it in a paragraph at the end of the discussion.
Considering the high drop out rate and the small sample size, I suggest to be cautious in your conclusions and to separate them in a specific paragraph at the end of the paper.
Best regards and good luck
Author Response
Dear authors,
your paper is valuable in its aims but some concerns have to be better addressed.
Methods have to be improved significantly.
Thank you very much for reviewing our paper. Please see our responses below to your helpful questions and comments.
In the methods section, a separate paragraph for statistical methods is required. You performed a pre/post study, but you should also specify which tests you used in case of parametric/non parametric variables.
Thank you for pointing this out. We separated out the statistics section. Our statistician has updated the methods section to reflect the tests that we used and how we addressed variables. This was necessary and we appreciate the recommendation.
Then, the design is not clear and it must be described both in the abstract and in the methods. Is it a retrospective study? Is it on a single cohort? Please specify and explain all these aspects.
Thank you for picking up on this missing information. This was a single cohort study. This has been added to the first paragraph of the methods section.
As regards the results, many concerns arise. A regression is required to assess if some factors could affect the results post to 1 year. Overall, it is useful to investigate other factors which could have affected results: for example, did participants carry out sporting activities? Also this factors can effectively influence physical function, body and mineral composition and so on. This aspect should be described in the discussion, in order to make it more objective and to remember that sport can also modify a disease perception by the patients. To do that, I suggest the following reference which explain how sport affect the perception of another disease by the patients:
Notarnicola A, Farì G, Maccagnano G, Riondino A, Covelli I, Bianchi FP, Tafuri S, Piazzolla A, Moretti B. Teenagers’ perceptions of their scoliotic curves. an observational study of comparison between sports people and non- sports people. Muscles Ligaments Tendons J [Internet]. 2019;9(2):225-35.
Thank you for this discussion. Overall numbers were too small to assess via a regression. However, this has been added to the limitations, improving this section. Regarding other activities, we considered engaging in sport as continued exercise. While some participants were exercising, none, unfortunately, were engaging in sports activities.
I suggest to separate the limitations of this study, specifically describing it in a paragraph at the end of the discussion.
Thank you for this suggestion. We have created a limitations paragraph before our summary/conclusions paragraph. We have also updated to include some of your comments from above.
Considering the high drop out rate and the small sample size, I suggest to be cautious in your conclusions and to separate them in a specific paragraph at the end of the paper.
Thank you for this comment. We have toned down our comments in the summary/conclusion paragraph at the end of the paper.
Best regards and good luck
Thank you again for reviewing our paper and your helpful comments.
Reviewer 2 Report
Comments and Suggestions for Authors
First, I would like to thank the authors for addressing the important issue of adherence to exercise therapy in breast cancer patients, which is a highly relevant topic, not only for researchers, but also for its significant value to patients.
This study aims to evaluate whether the enhanced physical and functional outcomes observed after a three-month resistance training program persist over one year. The manuscript is overall well written and adheres adequately to methodological requirements. However, some sections could be revised to provide more content, and the discussion could benefit from a broader analysis of the obtained results.
Please find below some detailed comments:
-
Correct the typographical mistake: "Abstract" should appear one line above.
-
The introduction presents several statements that require references (Lines 53–57 and 58–63).
-
Introduction: A critical aspect of strength training in cancer patients is program attendance and adherence to physical activity and healthy habits. The authors may consider investigating whether this general concern among cancer patients also applies specifically to women with breast cancer, and include a reference in the introduction.
-
Specify where the training sessions took place (e.g., gym facilities, hospital, home), and whether they were conducted individually or in groups.
-
The title of Figure 1 should be included.
-
If any participants required medication or medical procedures between the end of the resistance training intervention and the one-year assessment, this should be reported.
-
Although participants may not have engaged in formal physical activity, it would be relevant to mention whether they received physiotherapy (i.e lymphedema, low back pain). If this data is unavailable, the potential impact should be discussed in the results section.
-
The rate of adherence to exercise therapy is often influenced by clinical status and sociodemographic characteristics. The high adherence rate observed in your study may be related to such factors. I recommend discussing this point to strengthen the interpretation of your results. This article may be of interest in this context: Social inequalities in the use of physiotherapy in women diagnosed with breast cancer in Barcelona: DAMA cohort.
Author Response
Reviewer 2:
First, I would like to thank the authors for addressing the important issue of adherence to exercise therapy in breast cancer patients, which is a highly relevant topic, not only for researchers, but also for its significant value to patients.
This study aims to evaluate whether the enhanced physical and functional outcomes observed after a three-month resistance training program persist over one year. The manuscript is overall well written and adheres adequately to methodological requirements. However, some sections could be revised to provide more content, and the discussion could benefit from a broader analysis of the obtained results.
Thank you very much for reviewing our manuscript and the helpful comments.
Please find below some detailed comments:
- Correct the typographical mistake: "Abstract" should appear one line above.
Thank you for catching this. We have corrected it.
- The introduction presents several statements that require references (Lines 53–57 and 58–63).
Thank you for pointing this out. We have added references to support both of these statements.
- Introduction: A critical aspect of strength training in cancer patients is program attendance and adherence to physical activity and healthy habits. The authors may consider investigating whether this general concern among cancer patients also applies specifically to women with breast cancer, and include a reference in the introduction.
We have added several comments and references regarding exercise adherence in breast cancer patients. Thank you for this recommendation.
- Specify where the training sessions took place (e.g., gym facilities, hospital, home), and whether they were conducted individually or in groups.
We have specified that it was at the gym at our Exercise Oncology Center and in groups. This update has provided clarity in the Training Program section. Thank you.
- The title of Figure 1 should be included.
We have included this. Thank you for catching this.
- If any participants required medication or medical procedures between the end of the resistance training intervention and the one-year assessment, this should be reported.
This has been discussed in the results section and we have updated with specific injuries, listing them.
- Although participants may not have engaged in formal physical activity, it would be relevant to mention whether they received physiotherapy (i.e lymphedema, low back pain). If this data is unavailable, the potential impact should be discussed in the results section.
This is a good point. However, all of our breast cancer patients receive physical therapy and chronic follow up as part of their standard treatment, so we did not include this information due to confounding.
- The rate of adherence to exercise therapy is often influenced by clinical status and sociodemographic characteristics. The high adherence rate observed in your study may be related to such factors. I recommend discussing this point to strengthen the interpretation of your results. This article may be of interest in this context: Social inequalities in the use of physiotherapy in women diagnosed with breast cancer in Barcelona: DAMA cohort.
Our facility is located in an inner-city area around the poverty line. Our numbers are too small to show any relationship, but we have actually found the opposite of this study in an assessment of a large database from our center. However, this may have been because all participants were on study and thus exercise was provided without charge versus uncompensated exercise. We have added this in the discussion on adherence. Thank you for pointing this out.
Reviewer 3 Report
Comments and Suggestions for Authors
I suggest major revisions. Please see the PDF file.

Author Response
Reviewer 2:
Thank you for reviewing our manuscript.
Please reformulate the title
We have reformulated the title to the following: “Body composition benefits diminish one year after a resistance training regimen in breast cancer patients, but improvements in strength, balance, and mobility persist”
Add email
Thank you for pointing this out. The email is listed towards the end of the paper.
The introduction is rather short and does not provide enough information about the topic of the paper. Increase in fat tissue and decrease in muscle mass; the explanation of body composition should be the first paragraph in the INTRODUCTION (5-6 sentences).
We appreciate this recommendation and have enhanced and lengthened the introduction based on it. We feel this has improved the manuscript.
End of second paragraph add reference.
Thank you for catching this. We have added the reference here.
Third paragraph: Explain in more detail what resistance training is and what it entails. What are the recommendations for the application of resistance training in athletes, as well as in individuals diagnosed with cancer (duration, frequency, number of sessions, intensity)? List studies that have implemented resistance training and highlight the outcomes reported by the authors of those studies. Are you certain that resistance training is recommended for individuals with breast cancer, and which studies have applied it? Please provide detailed explanations.
Thank you for this suggestion. We have expanded this paragraph to include a more thorough discussion of resistance training, particularly in the breast cancer setting.
Therefore the aim of this study ..... Define the research hypothesis
We have added this to the last line of the introduction. Thank you for this suggestion.
Methods:
Separate the second paragraph to 2,2 Training Program
We have separated this as a header. Thank you.
Explain resistance training in a little more detail. How long was the introduction, main and final part. What exercises did the introductory and final part of the training entail?
We have added expanded this at length and added “that participants underwent a 2 week ramp up period utilizing the same exercises, but sets were gradually increased. Otherwise, we used the same exercises during the introduction as part of the exercise protocol.” This has provided clarity of the exercise regimen. Thank you for pointing this out.
Discussion
The first sentence in the discussion: The primary purpose of the study was to... After that, immediately confirm the hypothesis and explain the obtained results. Were the results in line with expectations or not?
Thank you for these recommendations. We have expanded the initial discussion as mentioned and systematically described the results of the protocol.
“Interestingly, the changes in strength, mobility, and balance were preserved regardless of whether an individual continued to exercise. “
Why? Please provide an explanation.
We have added that these changes were likely preserved due to neurological adaptations. Thank you for this suggestion.
“Intense resistance training utilizing heavy weights at full range of motion and at an adequate threshold can promote neurologic adaptation, which ultimately leads to improved strength, balance, and mobility.” Explain this sentence in more detail. For example, what are considered heavy loads? What percentage of 1RM? What is the recommendation for 1RM in individuals with breast cancer?
We have included a citation by Kraemer and loads lifted at 1-10 maximum repetition. There are few recommendations for breast cancer patients in terms of training intensity and 1RM. Our group hopes to provide data to lead to these recommendations in the future.
“These data are consistent with metanalyses revealing that several factors are associated with adherence, including contact with other participants, a supervised element, social support, and an exercise prescription.14” Explain the study.
We have expanded the discussion on this metanalysis in the discussion. Thank you for this recommendation.
4.1. Limitations and practical applications should be a separate subsection within the discussion. Also, clearly highlight the practical application of the obtained results. Who can benefit from these results in practice?
Limitations and practical applications were placed in a separate section. We expanded discussion on practical applications as well. Thank you for this recommendation.
- Conclusions should be a separate section.
We have separated this section as well.
The references are relevant to the topic of the paper. However, they are not presented in accordance with MDPI guidelines. Additionally, the paper contains only 17 references.
For example first reference: Caan, B.J.; Prado, C.M.; Meyerhardt, J.A.; Kroenke, C.H.; Weltzien, E.; Alexeeff, S.; Chen, W.Y.; Quesenberry, C.P.; Pantoja, D.; Sternfeld, B.; et al. Association of Muscle and Adiposity Measured by Computed Tomography With Survival in Patients With Nonmetastatic Breast Cancer. JAMA Oncol. 2018, 4, 798–804. https://doi.org/10.1001/jamaoncol.2018.0007.
Thank you for pointing this out. We have updated the citations to match MDPI guidelines and have added multiple citations during the editing process.
Round 2
Reviewer 1 Report
Comments and Suggestions for Authors
No futher corrections are needed
Author Response
Thank you for reviewing our paper for the Journal of Functional Morphology and Kinesiology.
Reviewer 3 Report
Comments and Suggestions for Authors
The authors have significantly improved the paper, which is now of much higher quality. I must also emphasize that the authors have addressed the comments and suggestions of the reviewer. In my opinion, the paper is ready for publication. I would only ask the authors to pay attention to the referencing style in the text. More precisely, the reference should be followed directly by brackets, for example [21]. whereas the authors have cited references as follows . [20]
This technical detail must be corrected.
Kind regards.
Author Response

(The authors gave the same response as above.)
